# Inhibition of Glucose Uptake Blocks Proliferation but Not Cytotoxic Activity of NK Cells

**DOI:** 10.3390/cells11213489

**Published:** 2022-11-03

**Authors:** Lea Katharina Picard, Elisabeth Littwitz-Salomon, Herbert Waldmann, Carsten Watzl

**Affiliations:** 1Department for Immunology, Leibniz Research Centre for Working Environment and Human Factors at TU Dortmund (IfADo), D-44139 Dortmund, Germany; 2Institute for Virology, University Hospital Essen, D-45147 Essen, Germany; 3Max Planck Institute of Molecular Physiology, D-44227 Dortmund, Germany; 4Chemical Biology, Faculty of Chemistry, Technical University Dortmund, D-44227 Dortmund, Germany

**Keywords:** natural killer cells, glucose transporter, metabolism, degranulation, cytokines

## Abstract

Tumor cells often have very high energy demands. Inhibition of glucose uptake is therefore a possible approach to limit the proliferation and survival of transformed cells. However, immune cells also require energy to initiate and to maintain anti-tumor immune reactions. Here, we investigate the effect of Glutor, an inhibitor of glucose transporters, on the function of human Natural Killer (NK) cells, which are important for the immunosurveillance of cancer. Glutor treatment effectively inhibits glycolysis in NK cells. However, acute treatment with the inhibitor has no effect on NK cell effector functions. Prolonged inhibition of glucose uptake by Glutor prevents the proliferation of NK cells, increases their pro-inflammatory regulatory function and reduces the stimulation-dependent production of IFN-γ. Interestingly, even after prolonged Glutor treatment NK cell cytotoxicity and serial killing activity were still intact, demonstrating that cytotoxic NK cell effector functions are remarkably robust against metabolic disturbances.

## 1. Introduction

Natural killer (NK) cells are innate lymphocytes that account for 5–15% of the mononuclear cells in peripheral blood. They play an important role in immune reactions against cancer and infections by killing transformed or infected cells [1]. NK cell cytotoxicity is induced via the engagement of activating receptors such as CD16, NKp30, NKG2D or 2B4 [2]. In addition, NK cells can produce cytokines and chemokines such as Interferon (IFN)-γ, tumor necrosis factor (TNF)-α or Interleukin (IL)-10, and they have an immunoregulatory function, whereby they can regulate cells of the innate and adaptive immune system [3]. In recent years it became clear that the activity of immune cells is greatly influenced by metabolism, as activities such as proliferation and effector functions are energy-demanding processes. Two major metabolic pathways are known to be involved in NK cell activation and proliferation: oxidative phosphorylation (OxPhos) and glycolysis [4]. Previous studies have shown that resting NK cells mostly use OxPhos, while upon activation they increase glycolysis and upregulate OxPhos [5]. This increases energy production and synthesis of building blocks required for biosynthetic processes to support proliferation and the production of effector molecules in NK cells [6,7]. For example, stimulation of NK cells by cytokines such as IL-12 and IL-15 induces their proliferation and leads to upregulation of OxPhos and glycolysis [5]. Similarly, Keppel et al. showed that stimulation by activating receptors is dependent on both glycolysis and OxPhos [8]. A major source of energy is glucose, but glutamine and fatty acids are also known to be important for cellular metabolism [9]. However, it is not yet known to what extent glutamine and fatty acids play a role in NK cells.

Glucose transporters (GLUTs) are 12 membrane-spanning receptors that allow glucose to enter the cell. In humans, 14 different GLUTs are known, and NK cells predominantly express GLUT-1 (*SLC2A1*) [10,11]. GLUT-1 and GLUT-3 are not only expressed by immune cells, but also in tumor cells due to their high metabolic demand and requirement of glucose for rapid proliferation. This makes glucose transporters a promising target for cancer therapies. Based on this fact, we have recently developed the GLUT inhibitor Glutor, which may be a promising candidate for potential anti-tumor therapies [12]. Glutor selectively inhibits GLUT-1, -2 and -3 and inhibits GLUT-3 more potently compared to GLUT-1. Treatment with Glutor can reduce proliferation and induces cell death in various tumor cell lines with an IC_50_ below 100 nM [12].

However, as activated NK cells may also rely on GLUT function for their anti-tumor activity, we explored whether Glutor affects the ability of NK cells to proliferate, kill target cells and secrete cytokines and chemokines, and whether NK cells change their phenotype in the absence of glucose availability. Therefore, we treated primary human NK cells with Glutor in short- and long-term experiments and analyzed their functions, phenotype and proliferation. Our results show that short-term treatment with Glutor has no effect on NK cell activities, whereas long-term treatment blocks NK cell proliferation and changes their phenotype, but only has a minor impact on NK cell cytotoxicity.

## 2. Methods

### 2.1. Isolation of NK Cells and Cell Culture

NK cells were isolated and cultured as described [13]. In short, human NK cells were isolated from peripheral blood mononuclear cells with the Dynabeads^®^ Untouched^TM^ Human NK Cell-Kit according to the manufacturer’s instructions (Invitrogen^TM^, Waltham, MA, USA). For experiments with resting NK cells, isolated NK cells were rested in IMDM medium (with GlutaMAX^TM^ by Gibco (Waltham, MA, USA), 10% FCS, 1% penicillin/streptomycin) overnight and then used for experiments. To generate pre-activated NK cells, isolated NK cells were seeded in 96-well round-bottom plates (Nunc, Thermo Fisher Scientific, Waltham, MA, USA) at a density of 1.5–2 × 10^6^ mL^−1^ with irradiated feeder cells (K562-mbIL15-41BBL) in medium with 200 U/mL IL-2 (National Institutes of Health Cytokine Repositor) and 100 ng/mL IL-21 (Miltenyi Biotec, Bergisch Gladbach, Germany). On day 8, NK cells were re-stimulated with fresh feeder cells. In the next weeks, NK cells were split on a density of 1.5–2 × 10^6^ mL^−1^ in the presence of 100 U/mL IL-2. On day 14 recombinant 2.5 ng/mL IL-15 (PAN Biotech, Aidenbach, Germany) was added, and after three weeks NK cells could be used as pre-activated NK cells. In case of long-term treatment, Glutor (100 nM) or DMSO (0.1%) were added to the culture and every 72 h new substance was added.

### 2.2. Cell Lines

K562 cells were cultured in DMEM (Gibco) with 10% FCS and 1% penicillin/streptomycin. MCF7 cells were cultured in DMEM (Gibco) with 10% FCS and 1% penicillin/streptomycin, 1% Sodium Pyruvat, 0.1% NeAA and 0.1% Insulin. HepG2 cells were cultured in DMEM (Gibco) with 10% FCS and 1% penicillin/streptomycin

### 2.3. RTCA Analysis

An amount of 50 µL of medium was added to the e-plates and background measurement was performed. Next, 3 × 10^4^ MCF7 cells or HepG2 cells in 100 µL medium per well were seeded and placed into the xCELLigence. Measurement was started for about 16 h, followed by the addition of pre-activated NK cells (3.75 × 10^3^) plus medium with DMSO (0.1%) or Glutor (100 nM) in 50 µL. Analysis was started and every 10 min for 72 h current was measured.

### 2.4. ^51^Chromium-Release Assay

Cytotoxicity was analyzed using a standard 4 h chromium-release assay as already described [13,14]. In short, target cells were labeled for 1 h at 37 °C and 5% CO_2_ on a Rotator with ^51^Cr. Afterwards, target cells were washed twice and added to the NK cells in a 96 v-well plate. NK cells were serial diluted (1:2) starting at an E:T 2:1. After 4 h incubation time, supernatant was collected and analyzed with the Wizard^2^ (Perkin Elmer, Waltham, MA, USA) gamma counter. The percentage of specific lysis was determined as follows:
(1)$$\frac{experimental\;release-spontaneous\;release}{maximal\;release-spontaneous\;release}\;\times \;100\%$$

### 2.5. Flow Cytometry

Degranulation assay: 0.1 × 10^6^ resting or pre-activated NK cells were pre-treated with Glutor (100 nM) or DMSO (0.1%) for 30 min and afterwards stimulated via plate-bound CD16-mAb (2 µg/mL), NKp30 (2 µg/mL), NKG2D + 2B4 (2 µg/mL) or control IgG (2 µg/mL) in the presence of anti-CD107a PE-Cy5 (1:50; Clone:H4A3; BioLegend, San Diego, CA, USA) for 2 h. CD107a- expression was measured on a BD LSRFortessa flow cytometer. Data were analyzed using FlowJo software (FlowJo, Ashland, OR, USA).

Intracellular staining: 0.2 × 10^6^ NK cells were stained for live/dead (zombie NIR 1:700; BioLegend) and CD56 using anti-CD56 BV421 (1:100; Clone:NCAM16.2; BD Biosciences, San Jose, CA, USA) for 15 min at RT. Intracellular staining was performed using 2% paraformaldehyde for fixation, Permeabilizing Solution 2 (BD Bioscience) for permeabilization and anti-GLUT1 (PE; 1:100; Clone:202915; R&D Systems, Minneapolis, MN, USA)), anti-pmTOR (PE-Cy7; 1:50; Clone:MRRBY; Thermo Fisher Scientific, Waltham, MA, USA) and anti-HK1 (AF647; 1:100; Clone: EPR10134(B); abcam) or Granzyme B (AF647; 1:200; Clone:GB11; BioLegend) and Perforin (FITC; 1:100; Clone:dG9; BioLegend) for intracellular staining (30 min, RT). Cells were measured on a BD LSRFortessa flow cytometer. Data were analyzed using FlowJo software (FlowJo).

Phenotype: Surface staining was performed using two different panels:

Panel 1: Zombie NIR (1:700; BioLegend), CD56 (BUV805; 1:4000; Clone:B159; BD Biosciences), CD3 (BUV563; 1:200; Clone:UCHT1; BD Biosciences), NKp46 (BV421; 1:50; Clone:9E2; BioLegend), 2B4 (FITC; 1:200; Clone:C1.7; BioLegend), NKp44 (PerCP-Cy5.5; 1:100; Clone:P44-8; BioLegend), DNAM-1 (AF647; 1:100; Clone:DX11; BD Biosciences), NKp30 (APC-Fire750; 1:100; Clone:P30-15; BioLegend), CD16 (PE-Dazzle; 1:100; Clone:3G8; BioLegend) and NKG2D (AF700; 1:100; Clone:FAB139N; R&D Systems).

Panel 2: CD38 (BUV395; 1:100; Clone:HB7; BD Biosciences), NKG2C (BUV496; 1:50; Clone:134591; BD Biosciences), CD3 (BUV563; 1:200; Clone:UCHT1; BD Biosciences), KLRB1 (BUV661; 1:100; Clone:NKR-3G10; BD Biosciences), CD56 (BUV805; 1:4000; Clone:B159; BD Biosciences), 4-1BB (BV421; 1:50; Clone:4B4-1; BioLegend), KLRG1 (BV510; 1:50; Clone:2F1; BioLegend), HLA-DR (BV605; 1:100; Clone:L243; BioLegend), FasL (BV650; 1:100; Clone:NOK-1, BD Biosciences), TIGIT (BV786; 1:100; Clone:741182; BD Biosciences), CD11a (AF488; 1:2000; Clone:HI111; BioLegend), CD8 (AF532;1:200; Clone:RPA-T8; Thermo Fisher Scientific), CD27 (PerCP-Cy5.5; 1:50; Clone:M-T271; BD Biosciences), NKG2A (PE; 1:100; Clone:Z199; Beckmann Coulter, Brea, CA, USA), CTLA-4 (PE-Dazzle; 1:100; Clone:BNI3; BioLegend), TRAIL (PE-Cy7; 1:50; Clone:N2B2; BioLegend), PD-1 (AF647; 1:400; Clone:EH12.1; BD Biosciences), CD18 (AF700; 1:800; Clone:TS1/18; BioLegend), Tim-3 (APC-Fire 750; 1:50; Clone:F38-2E2; BioLegend) and Zombie NIR (1:700, BioLegend). Cells were measured on a Cytek Aurora flow cytometer. Data were analyzed using FlowJo software (FlowJo).

### 2.6. SCENITH

An amount of 0.2 × 10^6^ NK cells/well were plated in 100 µL IMDM media and incubated in the presence or absence of Oligomycin (1 µM), 2-DG (100 nM) or Oligomycin + 2-DG for 30 min at 37 °C and 5% CO_2_. Next, 100 µL IMDM media containing Puromycin (10 µg/mL) was added to each well and incubated for 20 min at 37 °C and 5% CO_2_. Then, cells were washed with ice-cold PBS, and Fc-Block for 5 min at 4 °C was performed followed by a surface staining using zombie NIR (1:700; BioLegend), anti-CD56 BV421 (1:100; Clone:NCAM16.2; BD Biosciences) and anti-CD16 PE-Dazzle (1:100; Clone:3G8; BioLegend) for 30 min at 4 °C. Cells were subsequently washed, fixed and permeabilized using the Foxp3/Transcription Factor Staining Buffer Set (Invitrogen^TM^, Waltham, MA, USA) and then intracellular staining using anti-Puromycin AF488 (1:200; Clone:12D10; Sigma-Aldrich, St. Louis, MO, USA) and anti-pmTOR PE-Cy7 (1:100; Clone:MRRBY; Thermo Fisher Scientific) was performed for 30 min. Lastly, cells were washed again and then measured on a Cytek Aurora flow cytometer. Gylcolytic dependency, mitochondrial dependency, glycolytic capacity and FAO (fatty acid oxidation) and AAO (amino acid oxidation) capacity were calculated using the formula as already described by Argüello et al. [15]

### 2.7. ELISA

An amount of 0.1 × 10^6^ resting or pre-activated NK cells were pre-treated with Glutor (100 nM) or DMSO (0.1%) for 30 min and afterwards stimulated via plate-bound CD16-mAb (2 µg/mL), NKp30 (2 µg/mL), NKG2D + 2B4 (2 µg/mL) or control IgG (2 µg/mL) or via IL-12 (0.25 ng/µL) + IL-18 (1.25 ng/µL) as previously described [13]. After 16 h supernatants were collected. IFN-γ secretion was analyzed using the ELISA MAX™ Deluxe Set Human IFN-γ by BioLegend according to the manufacturer’s instructions. Measurement was performed using a GloMax^R^ instrument (Promega, Fitchburg, WI, USA).

### 2.8. Legendplex

Supernatants of long-term treated NK cells were collected 24 h after medium exchange and used for multiplex-bead array using the Legendplex Inflammation Panel 1 Human (BioLegend). Assay was performed according to the manufacturer’s instructions (BioLegend).

### 2.9. Migration Assay

Transwell plates (HTS Transwell^®^-96 Permeable Support with 5.0 µm Pore Polycarbonate Membrane, Corning, Corning, NY, USA) were used. Supernatants of long-term treated NK cells were collected 24 h after medium exchange. Then, 235 µL of the supernatants was added to the lower chamber and 0.25 × 10^6^ of freshly isolated PBMCs and granulocytes in 80 µL were added onto the insert. After 5 h of incubation (37 °C, 5% CO_2_), plates were incubated for 10 min at 4 °C, performed to induce detachment of monocytes. Cells were then removed from the lower chamber, transferred to a v-well plate, and pelleted by centrifugation. In the next step, cells were resuspended in 50 µL FACS-buffer containing an antibody mix of CD3 PerCP (1:100; Clone:SK7, BioLegend), CD56 BV421 (1:50; Clone:NCAM16.2; BD Biosciences), CD14 FITC (1:50 Clone: 61D3; Santa Cruz Biotechnology, Dallas, TX, USA), CD19 AF700 (1:50; Clone:HIB19; BioLegend), CD66b APC (1:200; Clone:G10F5; BioLegend) and CD11c PE (1:100; Clone: B-ly6; BD BioSciences) transferred to TruCount tubes (BD BioSciences) and incubated for 20 min at RT in the dark. Measurement of cells was performed using the BD LSRFortessa. Data were analyzed using FlowJo software (FlowJo) and absolute cell counts were calculated according to the manufacturer’s instructions.

### 2.10. Glucose Uptake Assay

An amount of 0.25 × 10^6^ long-term treated NK cells per well was seeded and Glucose uptake-Glo^TM^ Assay (Promega, Madison, WI, USA) was performed according to the manufacturer’s instructions. Luminescence was recorded using a GloMax^R^ instrument (Promega).

### 2.11. Serial Killing Assays

Assays were performed as described before [16]. In brief, microchips were equilibrated with 200 µL CTL medium for 1 h at 37 °C with 5% CO_2_, followed by removing the media and the addition of 200 µL CTL medium and 30 µL (30,000 cells) of resuspended MCF7 cells. After 5–10 min at 37 °C and 5% CO_2_, the distribution and amount of cells/well was checked, and if there were wells with 60–80 tumor cells, media were exchanged to stop sinking of further tumor cells. Microchip was incubated for 16 h at 37 °C and 5% CO_2_ to allow attachment of the tumor cells. After 16 h, a medium exchange was performed and 200 µL CTL medium containing the dead cell stain SYTOX Blue (1:1000, Thermo Fisher scientific) was added. Afterward, the microchip was placed into the incubation chamber of the ApoTome System with Axio Observer 7 microscope (Zeiss, Jena, Germany) equipped with a 20×/0.8 Plan-Apochromat objective and an incubation chamber with environmental control (37 °C, 5% CO_2_, humidity device PM S1). Next, long-term treated NK cells were added so that an optimal distribution was achieved with about 10–20 NK cells per well. Time-lapse microscopy was started and every 3 min for 16 h a picture was taken using an Axiocam 506 mono camera. SYTOX Blue was excited using the Colibri 7 LED-module 475 (filter set 38 HE LED) and brightfield was acquired using the TL LED module.

### 2.12. Seahorse Analysis

Energetic phenotype of pre-activated NK cells was analyzed using the glycolysis stress test kit (Agilent, Santa Clara, CA, USA). Experiment was performed according to the manufacturer’s instructions (Agilent). In brief, culture plates were coated with poly-l-lysine, followed by the addition of 0.25 × 10^6^ pre-activated NK cells in 80 µL RPMI medium without glucose per well. Cells were incubated for 30 min at 37 °C without CO_2_. In the meantime, injection ports were loaded with 10× concentrated substance (final concentration after injection: Port A: 100 nM Glutor/0.1% DMSO/medium; Port B: 10 mM Glucose; Port C: Oligomycin; Port D: 50 mM 2-DG) and subsequent calibration of the sensors was performed. Lastly, 100 µL RPMI medium without glucose was added to the cells and measurement was started.

### 2.13. RNA-Seq

The 10 × 10^6^ long-term treated NK cells (21 days) were used for RNA Isolation. RNA Isolation was performed using RNAeasy Mini Kit (Qiagen, Venlo, Netherlands) according to the manufacturer’s instructions with an additional step of DNAse treatment. Further analysis regarding purity and RNA sequencing was performed at the Genomics and Transcriptomics Facility (GTF) at the University Hospital Essen. In brief, QuantSeq 3′ mRNA-Seq Library Prep Kit FWD was used for library preparation. Sequencing was performed on the Illumina NextSeq500. Reads were processed with standard Illumina (San Diego, CA, USA) Basecalling, then trimmed with TrimGalore v0.6.0 and standard settings. Alignment was performed with hisat2 v2.2.1 to grch38 and standard settings. Raw counts were summed up with summarizeOverlaps from R-Package GenomicAlignments v1.8.4. RNA-Seq data were stored in the Gene expression omnibus archive (accession: GSE216156).

### 2.14. Statistics

Statistical analyses were performed using GraphPad Prism version 9 (Dotmatics, Boston, MA, USA).

## 3. Results

### 3.1. Glutor Changes the Energetic Phenotype of NK Cells

First, we determined whether NK cells are sensitive to the GLUT inhibitor Glutor. Therefore, we used the Seahorse technology to determine changes in the extracellular acidification rate (ECAR level) as a measure of glycolysis and changes in oxygen consumption rate (OCR level) as a measure of OxPhos in cultured human NK cells upon treatment of different concentrations of Glutor. We found a dose-dependent decrease in ECAR levels when administering 50 nM to 200 nM Glutor, while OCR levels increased with increasing concentrations of Glutor (Figure 1A). This demonstrates that GLUT inhibition by Glutor dose-dependently reduces glycolysis in NK cells, while it increases OxPhos to keep up with energy demands. Next, we investigated whether short-term Glutor treatment affects the NK cell metabolism and functions. Therefore, we treated resting and cultured human NK cells with 100 nM Glutor for 30 min before stimulation. Treatment with Glutor had no impact on NK cell degranulation or IFN-γ secretion induced via stimulation of the activating receptor CD16 (Figure 1B). Similarly, IFN-γ secretion upon stimulation via IL-12 and IL-18 was not affected by 30 min pre-treatment with Glutor (Figure 1C). Therefore, the functions of NK cells are not affected by short-term inhibition of glucose uptake.

### 3.2. Prolonged Inhibition of Glucose Uptake Does Not Affect NK-Mediated Tumor Cell Killing

As Glutor can affect the viability of different tumor cell lines [12] and as NK cells can directly kill tumor cells, we investigated both processes in parallel. Therefore, we used a real-time cell analyzer that determines the viability of cells using label-free impedance-based analysis. We analyzed MCF7 and HepG2 tumor cell lines, as MCF7 cells were shown to be sensitive to higher concentrations of Glutor, while HepG2 appeared to be unaffected by Glutor [12]. First, we seeded the tumor cells and measured the basal cell index. We then added pre-activated NK cells, Glutor or both together and continued to measure the cell index over 72 h (Appendix A). The analysis confirmed that MCF7 cells are sensitive to Glutor, as the cell index dropped significantly in the presence of the GLUT inhibitor (Figure 2A). The same could be observed upon addition of NK cells, demonstrating that the NK cells were killing the MCF7 cells. The combination of NK cells and Glutor resulted in an additive decrease in cell index, suggesting that both Glutor and NK cells were impacting the viability of MCF7 cells and that NK cell cytotoxicity was not negatively impacted by Glutor. In HepG2, we did not detect a significant decrease in cell index after the addition of Glutor, confirming that these cells are not sensitive to the effects of the GLUT inhibitor (Figure 2B). HepG2 cells were killed by NK cells, as the addition of NK cells decreased the cell index. The combination of Glutor and NK cells showed a comparable decrease in the cell index, again suggesting that Glutor does not significantly affect the killing activity of NK cells.

### 3.3. Long-Term Treatment with Glutor Results in an Altered Phenotype of NK Cells

To investigate the effect of Glutor on the proliferation of NK cells during long-term treatment, we used freshly isolated NK cells and expanded them with feeder cells in the presence of Glutor. Within the first week, an impaired proliferation of the Glutor-treated NK cells was already evident in direct comparison to the control. This continued in the following 2 weeks, resulting in a significant reduction in cell numbers (Figure 3A). Culturing of NK cells in glucose-free medium resulted in a similarly impaired proliferation within the first week, but in contrast to Glutor-treatment these NK cells did not survive (data not shown). Due to this strong effect of Glutor on proliferation, flow cytometry analysis of phosphorylated mTOR expression was performed at day 0, 2, 4 and 7, as pmTOR plays an important role during proliferation. Glutor treatment resulted in a delayed but prolonged upregulation of pmTOR, as levels were lower at day 4 but higher in Glutor-treated cells at day 7. The expression level of Hexokinase 1 (HK1), an essential enzyme for glucose metabolism, showed a similar trend (Figure 3B). After our 21-day expansion protocol the Glutor-treated cells had not expanded, but they were still viable. To analyze their phenotype, we measured 21 different surface molecules using spectral flow cytometry and observed significant changes between Glutor- and control-treated NK cells. Inhibition of GLUT-1,2,3 for three weeks lead to decreased expression of CD16, CD8, CD62L, TIGIT, 2B4, CD38, CD18 and NKG2C, while KLRB1, KLRG1 and HLA-DR were upregulated. We observed no changes in the expression of TRAIL, CD57, NKp30, NKp46 and CD11a, NKG2A, NKG2D, NKp44, DNAM-1 and 4-1BB (Figure 3C). Using intracellular staining, we detected higher expression levels of Granzyme B in Glutor-treated cells, whereas we saw a tendency towards lower expression levels of perforin (Figure 3D).

To confirm that Glutor was still inhibiting glucose uptake after long-term treatment we performed a glucose uptake assay on day 21, analyzing the uptake of 2-DG within one hour. Glutor-treated NK cells showed significantly reduced glucose uptake compared to the control (Figure 3E), confirming the effect of the inhibitor. To investigate the metabolic changes associated with glucose uptake inhibition we performed single-cell energetic metabolism by profiling translation inhibition (SCENITH). This is a flow-cytometry-based method for analyzing the metabolic profile of cells [15]. It allows conclusions to be reached about glucose and mitochondrial dependence as well as glycolytic capacity and fatty acid oxidation (FAO) in combination with amino acid oxidation (AAO). We observed a significant increase in mitochondrial dependency and a significant decrease in glycolytic capacity in Glutor-treated NK cells, while glucose dependency and fatty acid oxidation and amino acid oxidation capacity did not seem to be affected (Figure 3F). This confirms that in the absence of glucose uptake the cells changed their metabolism, which is likely the cause for the lack of proliferation and their phenotypic changes.

### 3.4. Long-Term Inhibition of Glucose Uptake Results in Upregulation of Glycolysis-Related Genes and Cell Cycle Arrest

To analyze these changes in an unbiased fashion, we performed bulk RNA sequencing analysis of long-term Glutor- or control-treated NK cells. We found a number of differentially regulated genes, and gene set enrichment analysis showed upregulation of genes related to glycolysis (e.g., *MXI1*), hypoxia (e.g., *ADM, CDKN1C*) and the p53 pathway (e.g., *PLK, GPX*) and reduced expression of genes related to oxidative phosphorylation (e.g., *ATP1B1, CPT1A*) in Glutor-treated NK cells in comparison to DMSO-treated controls (Figure 4A,B). This suggests that Glutor-treated cells try to counteract the lack of glucose uptake by upregulating glycolysis-related genes. However, the lack of glucose also seems to trigger cell cycle arrest and the expression of hypoxia-related genes.

As long-term treated NK cells can continue to take up small amounts of glucose (Figure 3E), we analyzed the potential upregulation of the various glucose transporters after long-term treatment. We examined the RNA-sequencing data set for *SLC2A1-14*, the genes encoding for the different glucose transporters, and found an increased expression of *SLC2A3*, *13* and *14* (Figure 4C) and increased surface expression of GLUT1 by FACS analysis (Appendix A). Volcano plot analysis showed significantly decreased gene expression of genes such as *CD38*, *CD7* and *death-associated protein kinase 2* (*DAPk2*) in Glutor-treated NK cells. In contrast, genes such as *BEN domain-containing 5* (*BEND5*), *cytotoxic t-lymphocyte antigen 4* (*CTLA4*), *nicotinamide phosphoribosyltransferase* (*NAMPT*) and *STC2* were upregulated upon Glutor treatment (Appendix A).

### 3.5. Functional Limitations Due to Long-Term Treatment with Glutor

Having established that the long-term inhibition of glucose uptake causes significant changes in the phenotype and metabolic profile of NK cells, we assessed the function of long-term treated NK cells. Feeder cell expansion and activation of NK cells increases their spontaneous secretion of cytokines and chemokines. Because of the phenotypical changes after long-term treatment, we took a closer look at this spontaneous secretion by testing the culture supernatant at day 21 of the expansion protocol by multiplex analysis. We found an increased spontaneous secretion of IFN-γ, TNF-α, IL-6, IL-8 and MCP-1, whereas there was a tendency toward lower IL-10 levels (Figure 5A). Based on these changes in the chemokine profile, we performed a Transwell migration assay. Freshly isolated granulocytes and PBMCs were put in the upper chamber of a Transwell plate and we measured the migration of these cells towards the lower chamber containing the supernatant of long-term Glutor- or control-treated NK cells. The chemokines in the supernatant of long-term Glutor-treated NK cells resulted in an increased migration of B cells, NK cells and NKT cells, while the migration of dendritic cells, granulocytes and monocytes was reduced (Figure 5B). This suggests that the regulatory activity of NK cells was altered by inhibiting glucose uptake.

Next, we analyzed the stimulation-dependent activities of the NK cells. Therefore, we stimulated NK cells with plate-bound antibodies against CD16, NKp30 and NKG2D + 2B4 and analyzed their degranulation by detecting CD107a expression via flow cytometry. Activation of NK cells via CD16 or NKp30 showed significantly fewer CD107a^+^ NK cells after long-term treatment with Glutor, whereas stimulation via NKG2D+2B4 resulted in comparable levels of CD107a^+^ NK cells (Figure 6A). Next, we analyzed IFN-γ secretion from long-term treated NK cells after 16 h of stimulation via plate-bound antibodies against CD16, NKp30 and NKG2D + 2B4 or via stimulation with the cytokines IL-12 + IL-18. Stimulation of Glutor-treated NK cells via CD16 or NKp30 showed a significantly decreased IFN-γ secretion, whereas NKG2D+2B4-stimulated NK cells displayed no changes. Long-term treatment with Glutor also led to significantly reduced IFN-γ secretion after stimulation via IL-12 + IL-18 (Figure 6B). This demonstrates that stimulation-dependent NK cell degranulation and cytokine production can be negatively impacted by inhibiting glucose uptake for long periods of time.

### 3.6. No Significant Change in the Serial Killing Activity of Long-Term Glutor-Treated NK Cells

The cytotoxic function of NK cells plays a central role in fighting cancer cells. While measuring CD107a detects the degranulation of NK cells, we also wanted to directly address if the NK-cell-mediated killing of tumor cells is influenced by Glutor. Therefore, we tested the cytotoxic activity of Glutor-treated NK cells in a ^51^chromium-release assay against K562 target cells. We observed no changes in the lysis of K562 cells between Glutor- and control-treated NK cells (Figure 7A). As NK cells have the ability to kill not only one tumor cell but several tumor cells within a few hours, inhibition of glucose uptake may influence the serial killing activity of the NK cells. To address this question, we performed a video microscopy-based serial killing assay using long-term treated NK cells against MCF7 tumor cells [16]. Using a 16 h incubation, we were able to distinguish between cytotoxic and non-cytotoxic contacts for up to six target cell contacts per individual NK cell. Glutor-treated NK cells only showed a slight reduction, but no significant difference when analyzing the number of cytotoxic contacts compared to control NK cells (Figure 7B). In addition, analyzing the percentage of serial killing NK cells (killing more than two target cells) and the number of kills per NK cell showed no significant differences between control and long-term Glutor-treated NK cells (Figure 7C). This demonstrates that long-term inhibition of glucose uptake did not have a significant impact on the serial killing activity of NK cells.

## 4. Discussion

Here, we characterized the effect of the GLUT inhibitor Glutor on the functional activities of human NK cells. Upon blocking glucose uptake by Glutor, NK cells immediately showed reduced glycolysis and increased OxPhos. However, these acute changes in their metabolism did not affect their functions, as NK cell cytotoxicity and IFN-γ production were not altered upon acute Glutor treatment. This could be due to the intracellular storage of glucose and possibly the use of alternative energy sources such as glutamine or fatty acids. Additionally, NK cell cytotoxicity relies mostly on pre-formed effector proteins that are stored within the lytic granules and that just need to be mobilized upon NK cell activation [17]. Therefore, only a little energy is needed for their immediate effector functions, and it has been described that the low basal metabolic rate of resting NK cells is sufficient to promote acute NK cell effector responses [8].

The stimulation of resting NK cells with feeder cells and cytokines induces NK cell proliferation and activation. This is accompanied by a strong increase in glycolytic activity to provide sufficient energy and precursors for anabolic processes [6,7]. In the presence of Glutor, glucose uptake and therefore glycolysis were severely inhibited in NK cells. This resulted in a lack of proliferation and was accompanied by a delayed induction of mTOR phosphorylation. Cytokines such as IL-2, IL-15 and IL-21 are known to relay their signals via mTOR, and mTOR plays an important role in the maturation, activation and metabolism of NK cells [18,19]. A previous study also showed a direct interaction between mTOR and HK1 [20], which could explain why we observed a delayed upregulation of HK1. In line with our previous results showing that Glutor treatment can induce an upregulation of GLUT-3 in tumor cells [12], our transcriptional analysis demonstrates that NK cells try to counteract the inhibited uptake of glucose by increasing the expression of *GLUT-3 (SLC2A3)*, *GLUT-13* (*SLC2A13*) and *GLUT-14 (SLC2A14)*, as well as other genes related to glycolysis. GLUT1 protein levels were also increased on Glutor-treated cells. Additionally, our metabolic SCENITH analysis also showed that Glutor-treated NK cells had significantly reduced glycolytic capacity, while showing higher mitochondrial dependency than control NK cells. The increase in hypoxia-related genes is likely a result of the lack of glucose in Glutor-treated NK cells, and the upregulation of genes related to the p53 pathway indicate that the NK cells are in cell cycle arrest, which would explain their lack of proliferation. Additionally, the upregulation of *CTLA-4* could play a role in the impaired IFN-γ production by long-term Glutor-treated NK cells [21]. While Glutor-treated NK cells did not proliferate, they still survived in culture and did not undergo apoptosis. This could be connected to the increase in *IL7R* expression, as IL-7 can promote the survival of NK cells [22]. Additionally, this may be because Glutor did not block glucose uptake completely or because the NK cells utilized alternative energy sources such as glutamine. In support of this, we observed that NK cells did not proliferate but underwent apoptosis when cultured in glucose-free medium or when we treated them with Glutor in combination with a glutaminase inhibitor (Appendix A).

This drastic change in metabolism of long-term Glutor-treated NK cells clearly had an impact on the phenotype of the cells. We observed a decreased expression of the activating receptors CD16, CD8, NKG2C and 2B4. In addition, CD38, which is known to play a role during activation and adhesion of NK cells, was downregulated [23]. Decreased expression could be observed for the adhesion molecule CD18 [24]. The reduced expression of these activating and adhesion receptors could also explain the reduced degranulation and IFN-γ production of the cells. The induction of the inhibitory receptor TIGIT [25] was diminished, which may also be a result of the impaired activation of Glutor-treated NK cells. The expressions of KLRB1, KLRG1 and HLA-DR were increased. KLRG1 is a marker for exhausted NK cells [26] and KLRB1 is a marker for NK cell function and primarily labels pro-inflammatory NK cells [27]. This is consistent with the changes in constitutive cytokine and chemokine secretion we observed in the Glutor-treated NK cells. The production of the pro-inflammatory cytokines IFN-γ, TNF-α and IL-6 was significantly increased, whereas the anti-inflammatory cytokine IL-10 showed reduced levels. For the chemokines MCP-1 and IL-8, we observed a marked increase in spontaneous secretion of both chemokines. Therefore, the inhibition of glucose uptake and glycolysis also changed the regulatory function of NK cells. This suggests that the increase in glycolysis is important for reducing the inflammatory phenotype of activated NK cells.

While the block in glycolysis completely prevented NK cell proliferation, the cytotoxic activity was only slightly reduced. Stimulation via CD16 or NKp30 showed a significant reduction in degranulation as well as IFN-γ secretion in Glutor-treated NK cells. However, stimulation via NKG2D + 2B4 did not show these functional impairments. This could be due to a different metabolic dependence of ITAM-based signaling which is mediated by CD16 and NKp30 versus the ITT and ITSM-based signaling that is initiated by NKG2D and 2B4 [2,28]. Another explanation could be the decreased expression of CD16 on the long-term treated NK cells; however, in the case of NKp30 we could not detect a significantly decreased expression. However, the reduced degranulation had no effect on the cytotoxic activity of NK cells in a 4h ^51^Cr release assay, and the slight reduction in the serial killing activity of these NK cells was not significant. Therefore, the NK cell cytotoxic machinery seems to be rather robust and only little affected by the altered metabolism upon prolonged inhibition of glucose uptake. Alternative energy sources such as glutamine could be sufficient to maintain the cytotoxic activity of NK cells. Additionally, the reliance on pre-formed effector molecules may explain the intact killing activity of Glutor-treated NK cells.

In summary, blocking glucose uptake by Glutor has a large impact on NK cell proliferation. The phenotype of the NK cells is changed by the lack of glycolysis, whereas the cytotoxic machinery seems to be rather robust and only shows a limited impairment upon long-term inhibition of glucose transporters. However, for tumor surveillance the lack of NK cell proliferation would have a large impact on NK-cell-mediated tumor control. Therefore, it is important to consider the effect on immune cells when targeting the metabolism of tumor cells.

## Figures and Tables

**Figure 1 cells-11-03489-f001:**
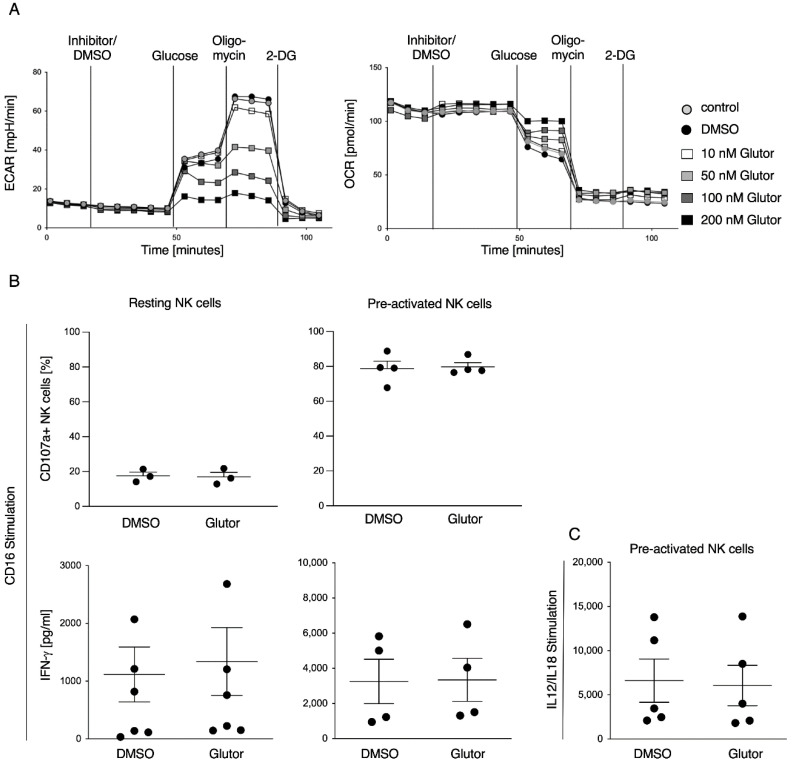
**Short-term inhibition of glucose transporters does not affect NK cell functions.** (**A**) Representative image of Seahorse analysis. ECAR levels and OCR levels of pre-activated NK cells before and after injection of medium, DMSO or Glutor (10 nM, 50 nM, 100 nM or 200 nM) followed by injections of glucose, Oligomycin and 2-DG: *n* = 3. (**B**) Resting NK cells or pre-activated NK cells were pre-treated with 100 nM Glutor or 0.1% DMSO followed by a stimulation via plate-bound CD16-mAb (2 µg/mL) for 2 h (CD107a assay) or 16 h (IFN-γ ELISA) in the continued presence of Glutor or DMSO: *n* = 3–6. Data were pooled from three or six independent experiments and each experiment was performed with one donor. Mean with SEM. (**C**) Pre-activated NK cells were pre-treated with 100 nM Glutor or 0.1% DMSO followed by a stimulation via IL-12 and IL-18 for 16 h (IFN-γ ELISA) in the continued presence of Glutor or DMSO: *n* = 5. Data were pooled from five independent experiments and each experiment was performed with one donor. Mean with SEM.

**Figure 2 cells-11-03489-f002:**
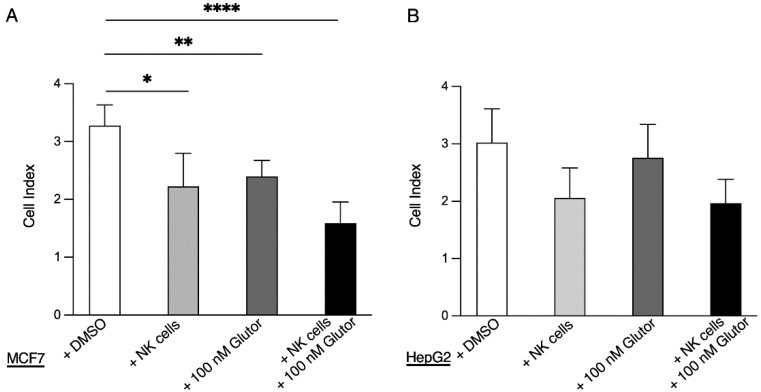
**The effect of Glutor on NK cells and tumor cells.** Impedance-based analysis of MCF7 cells (**A**) and HepG2 cells (**B**) in the presence or absence of 0.375 × 10^4^ pre-activated NK cells, 100 nM Glutor or 0.375 × 10^4^ pre-activated NK cells + 100 nM Glutor. The cell index at 72 h after addition of pre-activated NK cells +/− Glutor is shown: *n* = 3. Data were pooled from three independent experiments and each experiment was performed with one donor. Mean with SEM. Statistics: paired *t*-test. Significant differences are indicated by asterisk: * *p* ≤ 0.05, ** *p* ≤ 0.01, **** *p* ≤ 0.0001.

**Figure 3 cells-11-03489-f003:**
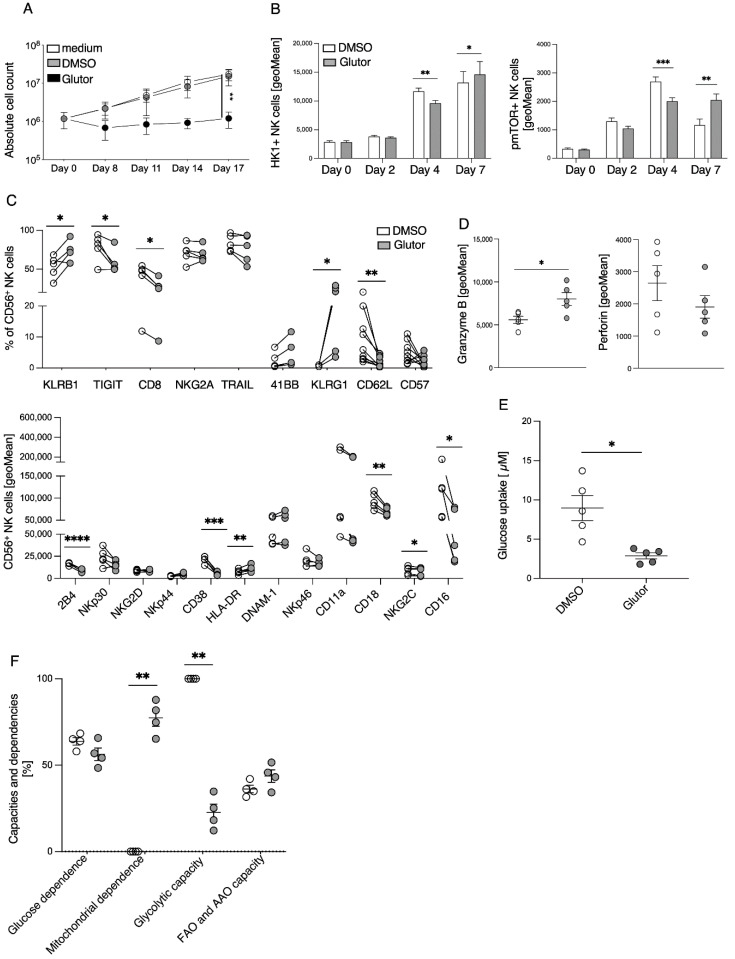
**Long-term treatment with Glutor results in an altered phenotype and in a reduced proliferation of NK cells.** (**A**) Proliferation of Glutor-treated (100 nM) in comparison to control-treated NK cells. Absolute cell count was calculated at indicated time points: *n* = 4. Data were pooled from four independent experiments and each experiment was performed with one donor. Mean with SEM. Statistics: paired *t*-test. (**B**) Expression level of HK1 and pmTOR on Glutor-treated (100 nM) and control NK cells was analyzed using flow cytometry at day 0, day 2, day 4 and day 7: *n* = 6. Data were pooled from four independent experiments and each experiment was performed with one donor. Mean with SEM. Statistics: paired *t*-test. (**C**) Expression level of several surface molecules was analyzed on day 21 of long-term treated NK cells (100 nM Glutor or 0.1% DMSO): *n* = 5. Data were pooled from five independent experiments and each experiment was performed with one donor. Statistics: paired *t*-test. (**D**) Granzyme B and perforin expression of long-term treated NK cells were analyzed using flow cytometry: *n* = 5. Data were pooled from five independent experiments and each experiment was performed with one donor. Mean with SEM. Statistics: paired *t*-test. (**E**) Glucose uptake assay of 100 nM Glutor-treated NK cells in comparison to control NK cells (0.1% DMSO): *n* = 5. Data were pooled from three independent experiments and each experiment was performed with one or two donors. Mean with SEM. Statistics: paired *t*-test. (**F**) Metabolic profile of long-term treated NK cells (100 nM Glutor or 0.1% DMSO) was analyzed using SCENITH: *n* = 5. Data were pooled from five independent experiments and each experiment was performed with one donor. Mean with SEM. Statistics: paired *t*-test. Significant differences are indicated by asterisk: * *p* ≤ 0.05, ** *p* ≤ 0.01, *** *p* ≤ 0.001, **** *p* ≤ 0.0001.

**Figure 4 cells-11-03489-f004:**
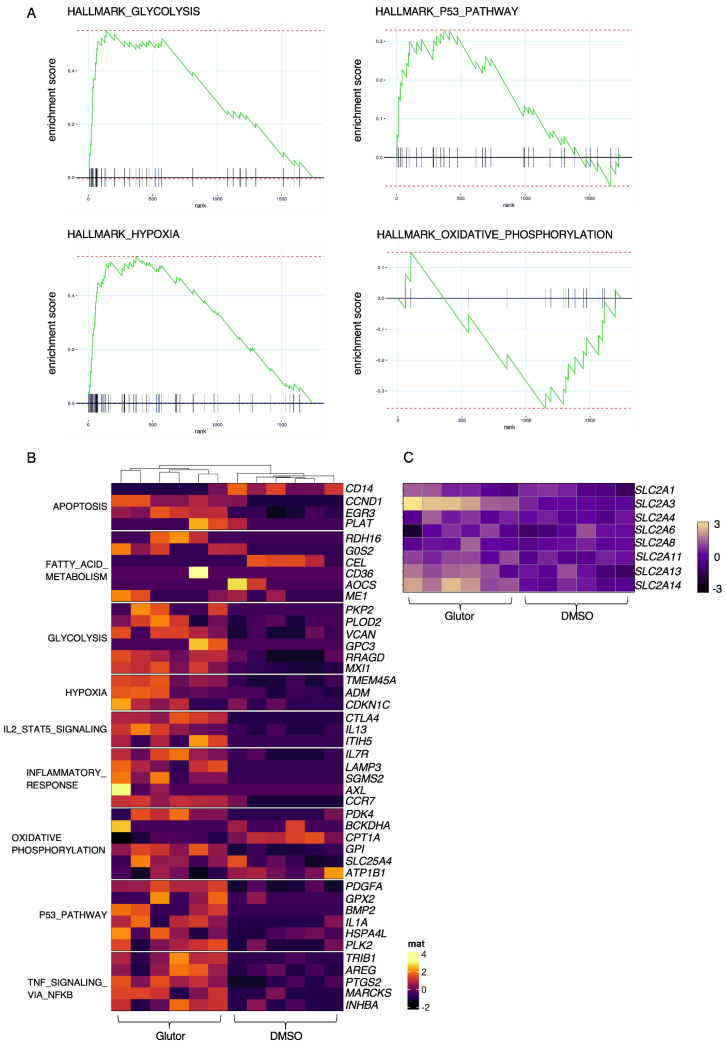
**RNA sequencing analysis of Glutor-treated NK cells.** (**A**) GSEA of long-term treated NK cells: *n* = 6. (**B**) Heatmap containing top 6 genes of significantly enriched hallmark gene sets: hallmark apoptosis, hallmark fatty acid metabolism, hallmark glycolysis, hallmark hypoxia, hallmark Il2 STAT5 signaling, hallmark inflammatory response, hallmark oxidative phosphorylation, hallmark p53 pathway, hallmark TNF-α signaling via NF-κB: *n* = 6. (**C**) Heatmap containing different SLC2A genes of long-term treated NK cells: *n* = 6.

**Figure 5 cells-11-03489-f005:**
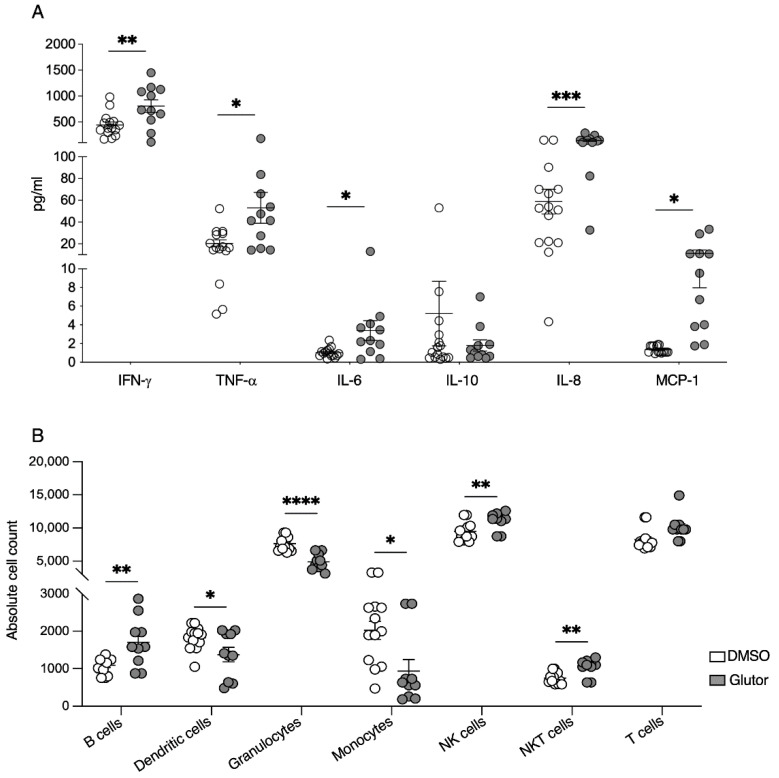
**Long-term treatment with Glutor leads to alterations in the chemokine profile.** (**A**) Supernatants of long-term treated NK cells (white: 0.1% DMSO; grey: 100 nM Glutor) were collected 24 h after splitting of cells and used for analysis of spontaneous secretion of different cytokines and chemokines using a cytometric bead-based array: *n* = 11–15. Data were pooled from eleven independent experiments and each experiment was performed with one or two donors. Mean with SEM. Statistics: paired *t*-test. (**B**) Absolute cell count of B cells, dendritic cells, granulocytes, monocytes, NK cells, NKT cells and T cells after performing a migration assay. Supernatants of long-term treated NK cells were (white: 0.1% DMSO; grey: 100 nM Glutor) used for the lower chamber and 0.25 × 10^6^ PBMCs + Granulocytes were added in the Transwell. After 5 h incubation, migrated cells were analyzed using flow cytometry: *n* = 10–13. Data were pooled from four independent experiments and each experiment was performed with 2–4 donors. Statistics: paired *t*-test. Significant differences are indicated by asterisk: * *p* ≤ 0.05, ** *p* ≤ 0.01, *** *p* ≤ 0.001, **** *p* ≤ 0.0001.

**Figure 6 cells-11-03489-f006:**
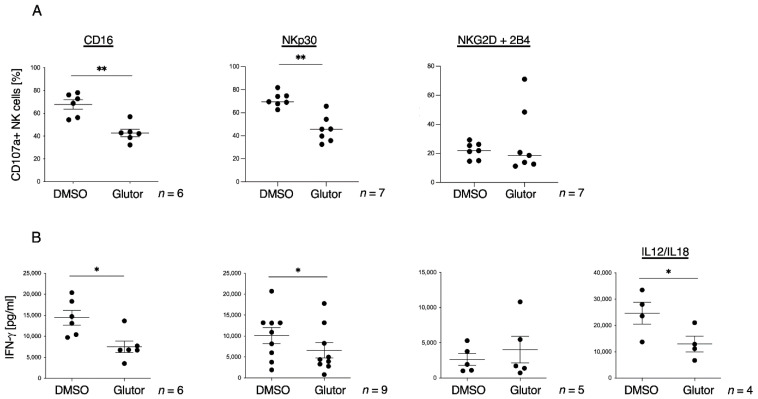
**NK cell functions are significantly diminished after long-term inhibition of glucose transporter.** (**A**) The 0.1 × 10^6^ long-term treated NK cells (100 nM Glutor or 0.1% DMSO) were stimulated via plate-bound antibodies against CD16, NKp30 or NKG2D + 2B4 for 2 h in the presence of an anti-CD107a antibody (PE-Cy5) in the continued presence of Glutor or DMSO followed by analysis of degranulation via flow cytometry: *n* = 6–7. Data were pooled from six or seven independent experiments and each experiment was performed with one donor. (**B**) The 0.1 × 10^6^ long-term treated NK cells (100 nM Glutor or 0.1% DMSO) were stimulated via plate-bound antibodies against CD16, NKp30 or NKG2D + 2B4 or via IL-12 + IL-18 for 16 h in the continued presence of Glutor or DMSO and IFN-γ secretion was measured (IFN-γ ELISA): *n* = 4–9. Mean with SEM. Data were pooled from four to nine independent experiments and each experiment was performed with one donor. Statistics: paired *t*-test. Significant differences are indicated by asterisk: * *p* ≤ 0.05, ** *p* ≤ 0.01.

**Figure 7 cells-11-03489-f007:**
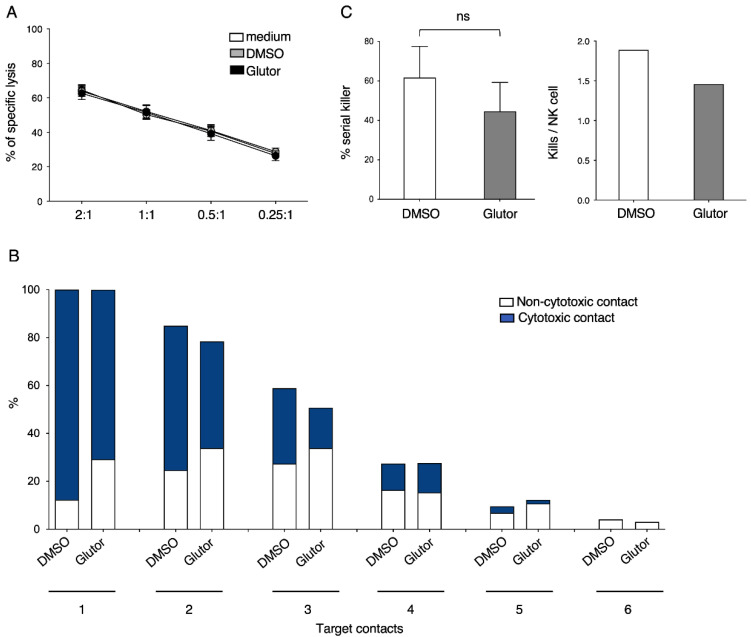
**Serial killing capacity is not affected by Glutor.** (**A**) Specific lysis of K562 using long-term treated NK cells (100 nM Glutor or 0.1% DMSO) in the continued presence of Glutor or DMSO was quantified using ^51^chromium-release assay starting at an E:T of 2:1: *n* = 3. Data were pooled from three experiments and each experiment was performed with one donor. Mean with SEM. (**B**) Serial killing assay of long-term treated NK cells (100 nM Glutor or 0.1% DMSO) as effector cells and MCF7 cells as target cells in the continued presence of Glutor or DMSO. Percentage of cytotoxic versus non-cytotoxic contacts were determined for each target cell contact (up to 6 target cell contacts were analyzed.) (**C**) Based on these data, kills per NK cell and percentage of serial killer were calculated. Between 50–70 NK cells were analyzed for each condition from 7 independent experiments. Each experiment was performed with one or two donors. For each donor, the percentage of serial killing NK cells was calculated. Mean and standard deviation is shown: ns, not significant.

## Data Availability

Raw data will be made available upon reasonable request to the corresponding author. RNA-Seq data were stored in the Gene expression omnibus archive (accession: GSE216156).

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
