# Peer review of "Inhibition of Glucose Uptake Blocks Proliferation but Not Cytotoxic Activity of NK Cells"

_cells, 2022, doi:10.3390/cells11213489_

Round 1

Reviewer 1 Report

The manuscript by Picard et al. focuses on investigating the effect of glucose uptake inhibition on human blood NK cell immune function. The manuscript is generally well-written and follows a logical structure. The authors provide interesting evidence that long-term inhibition of glucose uptake induced by Glutor treatment critically affects several essential NK cell immune functions, except for their cytotoxic capacity.

The results show that short-term treatment of human NK cells with Glutor does not affect CD107a expression, IFN-g production, or cytotoxic capacity. However, treatment of human NK cells with Glutor for 21 days critically affected their proliferation, changed their surface molecule phenotype by decreasing the expression of several NK cell activating receptors and adhesion molecules, and increased the expression of inhibitory molecules and Granzyme B expression. As expected, glucose uptake inhibition altered the NK cell metabolic profile, inducing the upregulation of glycolysis-related genes. Further, spontaneous production of IFN-g, TNF, IL-6, IL-8, and MCP-1 was increased following long-term treatment with Glutor, which seems to be associated with altered leukocyte migration of leukocytes treated with NK cell-derived supernatant. By contrast, long-term inhibition of glucose uptake of human NK cells decreased their stimulation-induced CD107a expression and IFN-g production. Surprisingly, long-term Glutor treatment did not affect the NK cell serial-killing ability. Although the authors provide interesting data showing the overall effect of Glutor treatment on human NK cell immune function, the manuscript would benefit significantly if the authors could address several issues.

Major comments: 

  1. Recent reports have shown that blood CD56bright and CD56dim human NK cells have distinct metabolic requirements at steady state and upon activation (Keating, J Immunol, 2016; Surace, Blood Adv, 2021). What effect does short-term and long-term Glutor treatment have on CD56bright and CD56dim NK cell effector functions?
  2. The heatmap containing the top 6 DEGs analyzed by RNA-seq shown in Fig. 4B should be reorganized, showing each gene set mentioned in the figure legend (OXPHOS, glycolysis, hypoxia, etc.). This would facilitate reading this figure.
  3. Among the DEGs that are increased after Glutor treatment in NK cells are several immune-related genes, including CD14, CTLA4, IL1A, CCR7, CD36, IL7R, AREG, LAMP3, and IL13. Interestingly, some of these genes are not typical NK cell-related genes. Is there any evidence in the literature that could relate the expression of these markers with the impaired NK cell function observed after Glutor treatment?
  4. Only one of the DEGs mentioned in the text (MXI1) appears on the heatmap showing the top 6 DEGs in Fig. 4B. What data did the authors use to conclude that the other genes listed in the text were differentially expressed in NK cells upon Glutor treatment? (lines 315-319).
  5. IL-8 is a known neutrophil and granulocyte chemotactic factor, while MCP-1 is a monocyte chemoattractant. What could explain the findings that although the authors observed increased levels of these chemokines in the supernatant of Glutor-treated NK cells, the migration of granulocytes and monocytes was decreased? Likewise, was the increased migration of B cells, dendritic cells, NK cells, and NK T cells caused by higher levels of cell-specific chemokines in the supernatant of Glutor-treated NK cells?
  6. How can the authors reconcile the opposite effects on IFN-g production in spontaneous (increased) versus stimulation (decreased) in long-term Glutor-treated NK cells?
  7. How can the authors explain their results that long-term Glutor treatment did not seem to affect NK cell cytotoxicity while showing increased Granzyme B expression, decreased CD107a expression, and a trend toward lower Perforin levels?
  8. The authors suggest NK cells switch to alternative energy sources such as glutamine or fatty acids upon long-term glucose uptake inhibition. These deductions are partly supported by their SCENITH and RNA-seq data. However, in line 449, they claim that Glutor-treated cells did not undergo apoptosis, and in line 453, they state that NK cells did not proliferate but underwent apoptosis when cultured in a glucose-free medium or with Glutor + glutaminase inhibitor. However, no viability/apoptosis data are not shown for either day 7 or day 21 in the manuscript to support these conclusions. This data should be included.
  9. The adverse effects of the long-term treatment with Glutor on CD107a expression, IFNg production, and NK cell phenotype are very compelling. It would be interesting to know whether these effects on NK cell function are temporary or long-lasting once glucose uptake is restored.

Minor comments: 

  • Line 18: It is not clear what the authors mean by saying that prolonged Glutor treatment of NK cells' increases their pro-inflammatory regulatory function' in the abstract. What data support this conclusion?
  • Line 20: Based on the overall data shown, the conclusion that 'NK cell effector functions are remarkably robust against metabolic disturbances' seems disproportionate, considering that the only effector function that appears to be unaffected in NK cells after prolonged glucose uptake inhibition is the NK cell cytotoxic capacity. This statement should be corrected.
  • Line 67: What is the purity range of the freshly isolated NK cells? And after the long-term culture?
  • Lines 93 and 138: The references format shows the author instead of its corresponding reference number.
  • Line 98: It is greatly appreciated that the authors show the antibody concentration they used for flow cytometry analysis. However, for reproducibility, it is equally important that they provide the clone number and vendor.
  • Line 108: anti-Glut1/PE is listed, but there is no graph showing Glut1 expression by flow cytometry.
  • Line 141: the concentration of IL-12/IL-18 used for cell stimulation is missing.
  • Line 202: The authors should include a brief paragraph in the methods detailing how the RNAseq library was generated and how the data were analyzed for Figure 4. Is the RNAseq library/database publicly available?
  • Line 345: The statement that the IL-10 levels in the supernatant of Glutor-treated NK cells were decreased is not adequately supported by the data shown in Fig. 5A; the statistical difference is missing.

  • Fig. 1A: The graph's circle symbol for "control" does not match the legend (in the graph is grey, whereas in the legend is white).
  • Fig. 2: Were these experiments done with fresh or pre-activated NK cells?
  • Fig. 2A: Is there a significant difference between in cell index when comparing the "+NK cells" and "+NK cells + Glutor" conditions?
  • Fig. 2B: Asterisks denoting statistical differences are missing between DMSO and "+NK cells" and "+NK cells + Glutor" conditions.
  • Fig. 4B, C: Gene names should be italicized.
  • Fig. 5: Panel labels are missing (A, B).
  • Fig. 5B: Is the migration of T cells statistically different between groups? 
  • Fig. 7C there seems to be a trend towards a decrease in % of serial killing, however since the authors are not including error bars nor are they showing individual points as in most of their figures, it is difficult to agree with their conclusion that there is no significant difference between DMSO and Glutor treatment.

Reviewer 2 Report

  The purpose of this study is to investigate the effect of Glutor, an inhibitor of glucose transporters, on NK cell response against cancer. Glutor inhibits glycolysis in NK cells and cancer cells.

  Although the scientific approach is logical. However, significant revision with a more correlation data analyze is needed and to be presented. Think about the real world of Glutor treatment, NK cell response against cancer in tumor microenvironment, and in correlation with data such as altered proliferation, phenotype, genes and chemokine profile under Glutor treatment.

1. The data on stimulation response (Figure 1B and 1C) in the presence of Glutor need to be analyzed and to be presented, in addition to resting NK cells or pre-activated NK cells response without Glutor. The time of stimulation response varies between 2h to 16hs, much longer than acute treatment of 30 minutes.

2. With the same logical, dada on NK cell function (Figure 6) and serial killing-capacity (Figure 7) in the presence of Glutor after long-term Glutor treatment need to be analyzed and to be presented.

Reviewer 3 Report

Review for cells-1976150-peer-review-v1, “Inhibition of glucose uptake blocks proliferation but not cytotoxic activity of NK cells”.

In the present study, the authors investigated the effect of Glutor, an inhibitor of glucose transporters, on the function of human Natural Killer (NK) cells, which are important for immunosurveillance of cancer. They showed that Glutor treatment effectively inhibits glycolysis in NK cells.

This study represents an impressive amount of work, is quite thorough, and the authors’ claims are well-substantiated based on multiple types of supporting evidence. The authors showed that although, short-term treatment with Glutor had no effect on NK cell activities, the authors found that prolonged inhibition of glucose uptake by Glutor prevents the proliferation of NK cells, increases their pro-inflammatory regulatory function, and reduces the stimulation-dependent production of IFN-γ. Interestingly, even after prolonged Glutor treatment NK cell cytotoxicity and serial killing activity were still intact, demonstrating that NK cell effector functions are remarkably robust against metabolic disturbances. The data within the paper are well-presented, clear, and thoroughly analyzed. However, there are some concerns with the lack of key experiments that decrease its scientific impact.

Major critiques: 

1.     Because NK cells predominantly express GLUT-1 (SLC2A1), it will be nice to see GLUT-1 protein expression following Glutor treatment.

2.     The authors have showed that upon blocking glucose uptake by Glutor, NK cells immediately showed reduced glycolysis and increased OxPhos. Which complex of the ETC do the authors think is primary affected? What does the RNA seq data suggest?

Minor critiques: 

1.     The figure number is missing for Figure 5 (A and B).

Figure S2: RNA sequencing is hard to read. The authors should enlarge the font size in the volcano plot, also nothing can be read for both X- and Y-axes.

Round 2

Reviewer 1 Report

The authors have satisfactorily addressed all my comments and suggestions, and the manuscript has improved significantly.

Reviewer 2 Report

Accept in the revised form.

Reviewer 3 Report

The authors have addressed to the reviewer's comments.